# Temporary Root Canal Obturation with a Calcium Hydroxide-Based Dressing: A Randomized Controlled Clinical Trial

**DOI:** 10.3390/antibiotics12121663

**Published:** 2023-11-26

**Authors:** Johannes-Simon Wenzler, Wolfgang Falk, Roland Frankenberger, Andreas Braun

**Affiliations:** 1Department of Operative Dentistry, Periodontology and Preventive Dentistry, Rheinisch-Westfälische Technische Hochschule University Hospital, Pauwelsstrasse 30, 52074 Aachen, Germany; anbraun@ukaachen.de; 2Department of Operative Dentistry, Endodontics and Pediatric Dentistry, Campus Marburg, University Medical Center Giessen and Marburg, Georg-Voigt-Strasse 3, 35039 Marburg, Germany; 3Center for Oro-Dental Microbiology, Hamburger Chausse 25, 24220 Flintbek, Germany

**Keywords:** calcium hydroxide, endodontics, bacterial reduction, medicaments, disinfection

## Abstract

Successful bacterial inactivation or elimination is essential for successful outcomes in endodontics. This study investigated the efficacy of a calcium hydroxide paste (Ca(OH)_2_) as a temporary medical dressing for 1 week after chemomechanical root canal treatment (CMRCT). Microbiological samples from 26 patients were collected after endodontic emergency treatment as follows: (1) removal of the provisional filling material; (2) CMRCT; (3) irrigation with sodium hypochlorite I (3%); (4) medicinal insertion of Ca(OH)_2_; and (5) irrigation with sodium hypochlorite II (3%). A microbiological examination was carried out after the specimens had been taken from the root canals via saline and sterile paper points. CMRCT resulted in a significant reduction in total bacterial load (TBL) in the root canal (*p* < 0.05). Additional irrigation (3) resulted in a further significant reduction in TBL (*p* < 0.05). In contrast, Ca(OH)_2_ medication did not prevent the bacterial load from returning to the previous level immediately after CMRCT, but did not increase above that level either (*p* < 0.05). However, the increase in TBL was significant (*p* < 0.05) in comparison with the disinfection groups (I/II). Administration of Ca(OH)_2_ for 1 week shows that in combination with an additional disinfection procedure, an increase in TBL must be expected, but not above the level of conditions after CMRCT.

## 1. Introduction

Infection of the pulp complex can have many causes. Caries, trauma, or periodontitis, for example, are all associated with colonization of the root canal system, primarily with bacteria and fungi from the oral microbiome. Ultimately, bacterial penetration in most cases unavoidably leads to inflammation of the pulp or the periapical complex. Treatment then involves initiating chemomechanical root canal treatment, including medicinal treatment [1]. This is because effective microbial elimination is essential as part of systematic endodontic therapy, as failures in root canal treatment can usually be attributed to insufficient microbial reduction. Root canal irrigation, as the first step, is an essential part of this stage of treatment, since mechanical preparation of the root canals alone does not usually lead to sufficient microbial elimination. In contrast, chemomechanical preparation with adjunctive disinfection methods can eliminate more than 92% of the microorganisms in the endodont [2,3]. In addition to conventional rinsing solutions such as sodium hypochlorite and other adjuvant disinfection methods such as ultrasonic or laser systems, medicinal pastes temporarily applied to the root canal also have an important role to play. Among other things, they are considered to have a germ-inactivating or germ-eliminating effect. One of these, calcium hydroxide, a widely used medication in endodontics, promises a bactericidal effect due to its high pH value and the release of hydroxyl ions, with only slight limitations on its effectiveness [4,5,6]. In vitro and in vivo studies have been published that demonstrate the effect in relation to different end points. Above all, microbiological tests—depending on the number of medicinal inserts and different previous disinfection protocols—show that calcium hydroxide pastes may be very effective [7,8]. In this context, other studies have underlined the importance of previously used disinfection measures in relation to residual bacteria in the root canal [9,10] and the resulting outcome [11]. However, microbial elimination never seems to be fully achieved, since even supplementary medicinal dressings with calcium hydroxide can only achieve almost complete sterility in 97% of cases, so that absolute sterility cannot be guaranteed in any case to date [12,13].

Against this background, the present in vivo study aimed to test a calcium hydroxide paste preparation on an aqueous basis with a pH of >12.5 and a calcium hydroxide content of 45% for its effectiveness in the context of root canal treatment. The study investigated the calcium hydroxide paste as a temporary root canal dressing after prior chemomechanical preparation in relation to the bacterial count reduction it achieved over a period of 1 week, testing the hypothesis that applying the dressing for a period of 1 week further reduces, or at least maintains at that level, the bacterial count achieved after disinfection.

## 2. Results

Table 1 and Figure 1 present the results for the microbiological and molecular biological samples collected (*n* = 26). At the start of the study (Baseline), a median value of 1.63 × 10^5^ CFU (min. 1.11 × 10^4^ max. 1.49 × 10^7^, IQR 1.60 × 10^5^) was observed.

However, a statistically significant difference was noted after chemomechanical root canal preparation (Root Canal Preparation), with a median value of 5.26 × 10^4^ CFU (min. 8.81 × 10^2^, max. 2.26 × 10^6^, IQR 1.14 × 10^5^; *p* < 0.05).

After subsequent disinfection in accordance with the study protocol, a further statistically significant reduction in bacteria was observed (Disinfection I), with a median value of 2.78 × 10^4^ CFU (min. 1.98 × 10^2^, max. 4.18 × 10^5^, IQR 3.05 × 10^4^; *p* < 0.05).

Temporary root canal obturation with a calcium hydroxide paste [Medication (7d)] showed a median value of 6.76 × 10^4^ CFU (min. 2.80 × 10^3^, max. 6.07 × 10^6^, IQR 7.78 × 10^4^) and did not prevent the bacterial count from rising back to the level immediately after chemomechanical treatment (Root Canal Treatment), although there was no increase above that level. However, there was a statistically significant difference from subsequent disinfection before temporary calcium hydroxide medication (*p* < 0.05).

The second disinfection measure (Disinfection II), with a median value of 1.68 × 10^4^ CFU (min. 1.89 × 10^2^, max. 1.71 × 10^6^, IQR 2.99 × 10^4^) before gutta-percha obturation, again led to a statistically significant reduction in the total bacterial load (TBL) in comparison with (Medication (7d)) (*p* < 0.05).

Individual bacterial species were detected in our study, but only inconsistently. After consultation with the microbiology laboratory, TBL was therefore used as the main parameter for analysis.

## 3. Materials and Methods

The present study was approved by the Ethics Committee (reference number 016/1749) and be carried out in full compliance with the current ethical principles (World Medical Association Declaration of Helsinki, version VI, 2002) at the Department of Operative Dentistry and Endodontology, University of Marburg, Germany. According to the diagnosis, clinical symptoms, and radiographic findings, teeth with suspected irreversible pulpitis were included, whereas teeth with suspected pulp necrosis were excluded. In addition, subjects had to be at least 18 years of age and had not received antibiotic treatment in the previous 6 months. Pregnancy, previous endodontic treatment of the tooth in concern, or a defect in the context of a periodontal-endodontic lesion were also considered as exclusion criteria. All 26 patients who fit the profile of the study according to the inclusion and exclusion criteria had previously given their consent to participate in the study.

### 3.1. Treatment Procedure

The initial endodontic emergency treatment of the study participants included, if necessary, restoration of the affected tooth with an adequate pre-endodontic composite restoration to ensure a suitable initial situation for further treatment. This then consisted of the isolation of the tooth with a rubber dam, the preparation of an access cavity, its irrigation with 3% sodium hypochlorite (5 mL), and an application of a calcium hydroxide paste (Calcicur; Voco GmbH, Cuxhaven, Germany). The tooth was then temporarily sealed using a foam pellet and a glass ionomer cement (Ketac Cem; 3M Espe, Seefeld, Germany). In the interests of patient welfare, no microbiological samples were taken at this point of the endodontic emergency treatment. One week after the emergency treatment, the patient returned for further treatment. In order to prevent bacterial contamination during the collection of the microbiological samples the tooth was again isolated with a rubber dam, which was then cleaned with a Lugol’s iodine solution (5%) and subsequently inactivated with sodium thiosulfate (5%; Dr. Franz Köhler Chemie GmbH, Bensheim, Germany). To disinfect the tooth surface hydrogen peroxide solution (30%; Carl Roth GmbH, Karlsruhe, Germany) was used. From this point onwards, microbiological samples were taken five times during the further treatment sequence according to the study protocol from study arm I. The root canals were therefore flooded with sterile saline solution for one minute and the microbiological samples were collected at each of the following time points using sterile paper points (ISO 30; VDW Antaeos GmbH, Munich, Germany) [2]:First sampling (Baseline)—following the removal of the temporary filling material and calcium hydroxide paste (Figure 2).Second sampling (Root Canal Treatment)—after clinical screening and inclusion of appropriate teeth into the study, the included teeth were prepared via chemomechanical root canal preparation up to size 30.09 (ProTaper Gold; Dentsply Sirona GmbH, Bensheim, Germany) under irrigation with sodium hypochlorite (3%; 5 mL total, applied over the duration of the root canal preparation) and ethylenediaminetetraacetic acid (15%; 2 mL).Third sampling (Disinfection I)—following additional rinsing with sodium hypochlorite (5 mL) (within the same treatment session as the second sampling).Fourth sampling [Medication (7d)]–following the removal of the temporary filling and calcium hydroxide paste and only if the tooth has been free of symptoms for 1 week.Fifth sampling (Disinfection II)—following a final rinse with sodium hypochlorite (5 mL) (within the same treatment session as 4th sampling).

Subsequently, each root canal was obturated with a gutta-percha filling and a root canal sealer, as well as adhesive sealing of the access cavity. 

All endodontic treatment procedures were performed according to a highly standardized protocol (study-internal standardization) to ensure comparability of the results, under the guidance of three dentists experienced in endodontics who were appointed as clinical investigators within the ethics approval as well as under the responsibility of the principal investigator responsible for the study.

### 3.2. Microbiological Analysis

An external laboratory (Oro-Dentale Mikrobiologie ODM, Kiel, Germany) performed the microbiological analysis of the previously collected samples. For this purpose, a quantitative real-time polymerase chain reaction (qPCR) was used and TBL was set as the main parameter for the analysis.

### 3.3. Preliminary Study 

In a further study arm, bacterial colonization of the root canals was investigated in accordance with the procedure described above, but without the inclusion of a canal dressing with calcium hydroxide paste between treatments. After four patients had been treated, an interim evaluation was carried out, with results showing that the TBL, with a median of 8.95 × 10^4^ CFU (min. 4.90 × 10^4^, max. 1.45 × 10^5^, IQR 3.68 × 10^4^), indicated an increase in the bacterial count in comparison with the condition immediately after chemomechanical root canal preparation, with a median of 1.56 × 10^4^ CFU (min. 1.36 × 10^4^, max. 3.26 × 10^4^, IQR 4.75 × 10^3^)—an even greater divergence of the TBL in comparison with the Disinfection I group, with a median of 4.50 × 10^3^ CFU (min. 1.80 × 10^3^, max. 7.30 × 10^3^, IQR 2.73 × 10^3^). Since the omission of interim root canal dressing was not in accordance with standard clinical practice and resulted in an increase in the bacterial load in the root canal, this study arm was discontinued; in addition, it had no further clinical relevance to the working hypothesis of reducing, or at least maintaining, the bacterial reduction achieved prior to tooth closure by applying a calcium hydroxide paste.

### 3.4. Statistical Analysis

To estimate the number of patients required, a power analysis was performed prior to the study [14]. The data measured and documented in an Excel spreadsheet (Excel 2016, Microsoft Office Professional Plus 2016, Microsoft Corporation, Redmond, WA, USA) were statistically analyzed using IBM SPSS Statistics for Windows, version 26.0 (Version: 29.0.0.0 (241), IBM Corporation, Armonk, NY, USA). The Shapiro-Wilk test was used as a test for the normal distribution of the measured values. As not all data were normally distributed, values were analyzed with a nonparametric test (Kruskal–Wallis). A pairwise comparison was performed with the Mann-Whitney test. Comparisons within each study group were performed with nonparametric tests for paired samples (Friedman and Wilcoxon test). When multiple statistical tests were performed simultaneously on a single data set, Bonferroni correction of the critical p-value was applied. At *p* < 0.05, differences were deemed statistically significant. Boxplot diagrams display the minimum and maximum values (whiskers), as well as the median and first and third quartiles. Asterisks are used to identify outlier values, which are defined as values that are more than 1.5 to 3 times the interquartile range (IQR).

## 4. Discussion

The medicament most commonly used for temporary intracanal dressings is calcium hydroxide, which has a strong antibacterial effect due to its alkaline pH (12.5–12.8) and which is attributed to a resulting reduction of the intracanal bacterial load [6,15,16]. It is also assumed to have an influence on the outer root surface in terms of periodontal recovery [1,17,18], to be effective also against bacterial products such as lipopolysaccharides [19] and to control inflammatory exudates from the periapical area [20]. The results of the present study are largely consistent with previous findings regarding the basic assumptions about calcium hydroxide. In contrast to many previous studies, however, the results of the present study show that the total number of bacteria in the root canal increased significantly compared to the previous additional root canal irrigation (Disinfection I) despite the use of calcium hydroxide as a medicinal insert. Since the antibacterial efficacy of calcium hydroxide has previously been demonstrated in in vitro studies [4,21], the question remains as to how this renewed increase in bacterial colonization might be explained here. Considering that in vitro models of any kind can never reflect the complexity of actual in vivo situations, it must be taken into account that in vitro studies on antibacterial efficacy are often conducted using bacterial suspensions or artificial biofilms consisting of only a few bacterial species [22,23,24,25]. Therefore, other factors, such as additional protective properties of well-established biofilms may be of decisive importance here. These properties include a biofilm matrix, an altered growth rate of biofilm organisms, as well as other physiological changes [26,27]. Thus, most bacteria also showed increased resistance to alkaline challenges/stress when organized in the biofilms [26,27,28]. 

A possibly insufficient coating of the root canal wall with calcium hydroxide must also be considered here as a possible reason for the failure of the medicinal insert and the observed increase in TBL. A potential impairment of the coating of the root canal walls is the so-called vapor lock effect, which is considered a key limitation in the disinfection of root canals with rinsing solutions. This term refers to gas accumulations that usually occur in the lower third of the root canal (due to anatomical, physical, or chemical influences) and prevent the deeper penetration of rinsing solutions or medicinal pastes as well as their homogeneous distribution and thus also prevent their optimal effectiveness (e.g., due to lack of contact with the inner surfaces or dentinal tubules of the root canal) [29,30,31]. A recent study by Puleio et al. from 2023 investigated the vapor lock phenomenon during endodontic treatment using the CBCT technique and demonstrated its presence in almost all endodontic treatments, especially in the apical canal third [32]. With regard to the present study, it cannot be ruled out that the calcium hydroxide pastes used for medicinal inserts are also subjected to the phenomenon of the vapor-lock effect, thus impairing the sufficient coating of the entire surface of the root canal area. Furthermore, the ideal time that calcium hydroxide must be present in the root canal in order to comprehensively disinfect the canal system is not yet known. Nor is it known to which extent the type of bacteria as well as their location in the root canal influence the result [33]. Nevertheless, previous studies have already shown that up to 25% of bacteria can remain within the root canal, which is consistent with the results of the present [34,35,36]. As a result, remaining bacteria within the root canal could proliferate despite the use of a medicinal insert and thus cause a renewed increase in the TBL value at the time of the fourth sampling [Medication]. 

To date, only a few studies are available on the effectiveness of calcium hydroxide as a medicinal insert in vivo. Most of these studies primarily focused on treatment methodology and investigated single versus multiple-visit approaches in terms of disinfection and medicinal inserts. Nevertheless, interesting data concerning the question addressed in the present study can be obtained. An in vivo study on teeth with apical periodontitis by Vera et al. showed the importance of temporary medication of root canal–treated teeth from a histobacteriological point of view. In comparison with the so-called one-visit group (chemomechanical root canal treatment and obturation during the same appointment), the two-visit group (additional medication with calcium hydroxide paste for 1 week before obturation) showed an improved microbiological status. The study confirms the value of using calcium hydroxide paste for additional bacterial reduction and may provide an initial explanation for the rebound in total bacterial load observed in the present study as involving apparently surviving residual bacteria. The latter were found more frequently and in greater abundance in ramifications, isthmuses, and dentinal tubules—thus showing additional dependence of the results on the individual root canal anatomy [7]. In contrast to Vera et al. and the present study, Kvist et al. used bacterial cultures as evidence for bacterial reduction. The study also compared a one-visit group and a two-visit group with each other. The authors found that there were no statistically significant differences between the two study groups [8,15]. Unfortunately, closer comparison with the present study is not possible, as the study protocol used by Kvist et al. did not take samples between the individual disinfection procedures. However, from the results reported between the one-visit and two-visit groups, a presumed trend toward greater recolonization with bacteria can be interpreted—although, of course, in contrast to the present study, a potentially higher bacterial load due to the diagnosis of apical periodontitis should also be considered. The complexity of microbiological sampling and factors influencing it—such as the anatomy of the root canal system in combination with temporary root canal pastes—must be particularly emphasized at this point [7,15,37]. The present study shows a high level of diagnostic accuracy in comparison with other approaches, free from the influence of calcium hydroxide. In contrast, it even shows that despite the temporary use of the substance, a transient increase in the total bacterial load is detectable—i.e., it does not lead to false-negative results [15,34]. Regardless of the initial situation and the treatment method, the situation in the current study shows a rather heterogeneous picture. Multiple application of calcium hydroxide is preferable to a single visit, and a storage time of 7–45 d appears to provide advantages [1,38,39]. It is possible that a longer calcium hydroxide storage period would also have shown a greater effect on bacterial reduction in this study. A new approach could possibly prove this.

In this context, however, attention should also be drawn to the actual disinfection protocols, i.e., to the use of disinfectant rinsing solutions. Even in the two disinfection groups (Disinfection I and II) in the present study, the significant impact of additional sodium hypochlorite irrigation on the total bacterial load can be seen. It is generally known that rinsing solutions such as sodium hypochlorite and chlorhexidine reduce the total bacterial load by up to 95% [14,16]. With regard to the efficacy of rinsing solutions alone, this is also confirmed by numerous publications and the resulting meta-analyses and systematic reviews of clinical trials, although it should be noted that the results are limited due to inconsistencies between articles and the lack of clinically relevant results, among other factors [40,41,42]. It is therefore not surprising that the two disinfection methods showed better disinfection efficacy in relation to the TBL, and a rebound can also be explained due to the limited effectiveness of the calcium hydroxide [43]. The studies by Siqueira et al. and Vianna et al. partly support the present results, but of course without the additional disinfection measures included in our protocol. Both studies investigate the combination of sodium hypochlorite and calcium hydroxide during root canal treatment, looking at the bactericidal effect. The results show that significant bacterial reduction can be expected after chemomechanical root canal preparation with sodium hypochlorite, but statistically significant differences were not found between root canal treatment and medicinal calcium hydroxide insertion (for 7 d). An additional significant bacteria-reducing effect by calcium hydroxide was therefore not confirmed [9,44]. This is also in line with our findings when comparing the study groups. Nonetheless, compared with the present data, it can be concluded that an additional disinfection measure during root canal treatment is essential in order to additionally reduce the TBL even after medication for 7 d.

Another important issue is that the increase in the TBL between the additional disinfection procedure [Disinfection I] and drug treatment after 7 days [Medication] could be due to presumed coronal leakage or to the skills of the operators in this study, in addition to the limited efficacy of calcium hydroxide. Coronal leakage, i.e., gaps between restorative materials and the cavity wall, caused by inadequate temporary fillings during or after endodontic therapy, may be responsible for bacterial recolonization and thus an increase in the TBL. In the present study protocol, the teeth were sealed using calcium hydroxide, a foam pellet, and a glass ionomer cement for 7 d. Recontamination can, of course, hardly be excluded. Several studies have shown advantages, especially through combinations of different materials or with self-adhesive products, but penetration tests using dye, bacteria, or glucose, for example, have never confirmed the complete impermeability of the different filling materials. In some cases, recontamination was detected within 48 h [45,46,47]. This could also explain a recurrent increase in the TBL between the (Disinfection I) and (Medication) groups in the present study, along with differences in the skills of the operators mentioned above. As different dentists were responsible for patient care in this study, the results are always dependent on the dentist concerned. For example, in the area of chemomechanical treatment, disinfection, calcium hydroxide application, calcium hydroxide removal, and temporary fillings mentioned above, variations can occur that naturally affect the results.

The use of the qPCR method and the omission of methods such as bacterial culture is also a factor worthy of discussion, which has already been mentioned. As in the study by Wenzler et al. [2], this study targeted the total load of bacteria previously defined as endodontically relevant, with the advantage that the qPCR method could be used specifically. Of course, as with the culture method, the opportunity was taken to evaluate the results openly—i.e., to detect bacteria not considered, or a variation in the spectrum. However, the PCR method has an advantage over the culture method for detecting difficult species, bacteria in a viable but nonculturable state, and species that have not yet been cultured. The problem with qPCR is certainly that dead cells or extracellular DNA can also lead to errors in the results due to amplification and detection [14,48,49]. However, this problem could be avoided in future studies by adding propidium monoazide (PMA), which can be used to separate dead from live bacteria [50]. In this study, qPCR demonstrated its advantages due to its sensitivity and specificity, as well as the time advantage in comparison with the culture method—also confirmed by the studies mentioned above. The qPCR method was considered suitable for this study.

With regard to the clinical significance of the present study, it should be noted that the observation period of the calcium hydroxide insert is limited to only one week and that no date on the overall (clinical and radiological) success of the endodontic treatments was included. Particularly with regard to the clinically relevant long-term success of endodontic treatments, it would be interesting to evaluate the follow-up of the patients treated in the scope of this study–possibly by means of a retrospective study. Nevertheless, within the limitations of this study, the results may support previous assumptions that successful root canal treatment depends on keeping the bacterial count below a threshold that the immune system can cope with. The long-term success of root canal therapy, however, cannot always be guaranteed due to a resurgence of bacteria in cases of immune suppression brought on by illness or, for example, aging [51]. Endodontic therapy should therefore always focus on the greatest possible reduction in bacteria. In addition to chemomechanical preparation and adjuvant disinfection protocols, which are generally and still considered to be the most important step in root canal disinfection, the introduction of intracanal medications such as calcium hydroxide is considered necessary in order to keep the bactericidal effects of the irrigation solutions constant and in addition to achieving maximum eradication of pathogens from the root canal [9,15,42,52].

The present study intentionally omitted an additional control group without calcium hydroxide placement (see Section 3.3 above, Preliminary Study). On the one hand, the calcium hydroxide dressing corresponds to the standard procedure for endodontic treatment in the clinic, and on the other hand, only the hypothesis of an additional reduction of the bacterial load in the root canal by the calcium hydroxide dressing over a period of 1 week was to be tested. This hypothesis was clearly rejected in the present study design. Further studies should follow, investigating the efficacy of other medicinal deposits that can effectively reduce the bacterial load in the temporarily closed root canal or at least maintain it at the level prior to closure.

## 5. Conclusions

Chemomechanical root canal treatment with a 3% sodium hypochlorite rinsing solution significantly reduced the bacterial count in the root canal, as did additional rinsing with sodium hypochlorite after the root canal treatment. Temporary administration of calcium hydroxide-based root canal medication for 7 d did not prevent the bacterial count from returning to the level immediately after chemomechanical treatment. The study shows that bacterial recolonization should be expected when a calcium hydroxide paste is used for temporary medicinal treatment of the root canal. However, the recolonization can be significantly reduced again by additional rinsing with sodium hypochlorite. Modern endodontics appears to be continuing to rely on the paradigm that disinfection with irrigation solutions rather than temporary root canal medication remains the essential element in endodontic therapy.

## Figures and Tables

**Figure 1 antibiotics-12-01663-f001:**
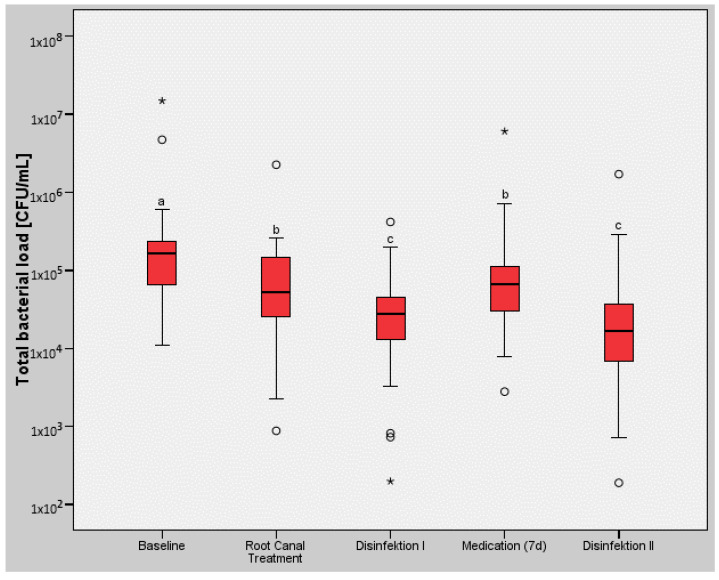
Box plot diagram for the total bacterial load (TBL) in the different groups; the same lower-case letters indicate no statistically significant differences (*p* > 0.05) between groups; mild outliners: ° and extreme outliners: *.

**Figure 2 antibiotics-12-01663-f002:**
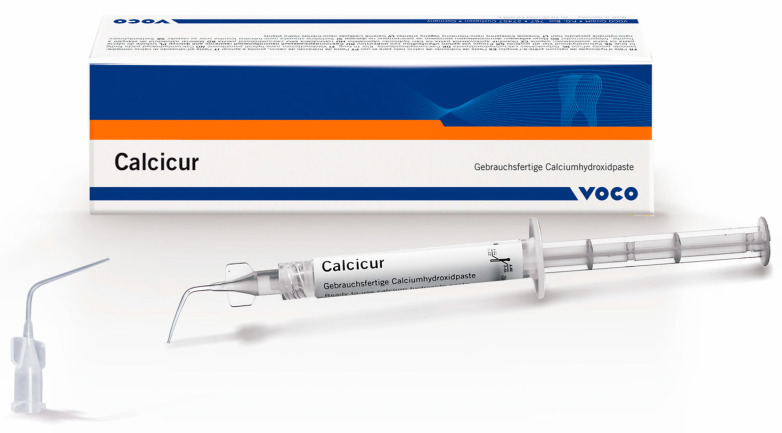
The calcium hydroxide paste used for temporary insertion into the root canal (Calcicur; Voco GmbH, Cuxhaven, Germany).

**Table 1 antibiotics-12-01663-t001:** Total bacterial load in CFU/mL in the study groups at the different time points for sample collection.

	Baseline	Root Canal Treatment	Disinfection I	Medication (7d)	Disinfection II
Mean	9.05 × 10^5^	1.65 × 10^5^	5.12 × 10^4^	3.34 × 10^5^	1.02 × 10^5^
Standard Deviation	2.99 × 10^6^	4.33 × 10^5^	8.53 × 10^4^	1.18 × 10^6^	3.35 × 10^5^
Median	1.63 × 10^5^	5.26 × 10^4^	2.78 × 10^4^	6.76 × 10^4^	1.68 × 10^4^
Minimum	1.11 × 10^4^	8.81 × 10^2^	1.98 × 10^2^	2.80 × 10^3^	1.89 × 10^2^
Maximum	1.49 × 10^7^	2.26 × 10^6^	4.18 × 10^5^	6.07 × 10^6^	1.71 × 10^6^
Interquartile Range	1.60 × 10^5^	1.14 × 10^5^	3.05 × 10^4^	7.78 × 10^4^	2.99 × 10^4^
*n*	26	26	26	26	26

## Data Availability

The data presented in this study are available upon request from the senior author, A.B.

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
