# Peer review of "Temporary Root Canal Obturation with a Calcium Hydroxide-Based Dressing: A Randomized Controlled Clinical Trial"

_antibiotics, 2023, doi:10.3390/antibiotics12121663_

Round 1
Reviewer 1 Report
Comments and Suggestions for Authors
Thank you for giving me this opportunity to review the article entitled "Temporary root canal obturation with a calcium hydroxide-based dressing: a randomized controlled clinical trial". I carefully reviewed the submitted set of the manuscript and I would like to clarify some doubts.
1. when appears "in vitro" and "in vivo" in the manuscript, it must be italicized;
2. The authors put "2. Results" before the "3. Methods";
3. Item "2.2. Treatment Procedure ": the authors need to explain much better the protocol because its to hard for me understand. For example, I was unable to understand whether a sample was taken or not during the emergency appointment.
4. In this first emergency endodontic treatment, authors should clarify what type of temporary restoration is used (although it is written in the discussion);
5. In the same first emergency endodontic treatment, did the teeth have all four walls necessary to start the endodontic treatment? If not, was pre-endodontic restoration performed before starting treatment?
6. "2nd sampling (Root Canal Treatment)": all the root canals was instrumented with the same final file (30.09)? Even though they are molars or single-rooted teeth?
7. How many different operators performed the treatments? Were they calibrated with each other? How was the calibration done?
8. In relation to the "Discussion" section, the authors need to modify a little bit. They should focus more on the results obtained and interpret them according to the existing scientific evidence. There is no need to evaluate topics that they have not analyzed (f. ex. single appointment vs multiple appointment). Furthermore, the discussion should focus more on calcium hydroxide (objective of the study) and not so much on irrigation with sodium hypochlorite.
Moderate editing of English language required
Author Response
Dear colleague,
thank you very much for your thorough review of our manuscript.
Please find enclosed the revised version of our manuscript entitled "Temporary root canal obturation with a calcium hydroxide-based dressing: a randomized controlled clinical trial", which has been approved by all authors.
Comments and Suggestions for Authors:
Reviewer: When appears "in vitro" and "in vivo" in the manuscript, it must be italicized;
Answer authors: We thank you for your time, your effort and your comments. We have incorporated this point into the manuscript.
Reviewer: The authors put "2. Results" before the "3. Methods";
Answer authors: Unfortunately, this point is due to the magazine's specifications. The team of authors themselves find this difficult, as it significantly influences the flow of reading.
Reviewer: Item "2.2. Treatment Procedure ": the authors need to explain much better the protocol because its to hard for me understand. For example, I was unable to understand whether a sample was taken or not during the emergency appointment.
Answer authors: The reviewer is right. Therefore, the team of authors has improved the text as follows:
“The initial endodontic emergency treatment of the study participants included, if necessary, restoration of the affected tooth with an adequate pre-endodontic composite restoration to ensure a suitable initial situation for the further treatment. This then consisted of the isolation of the tooth with a rubber dam, the preparation of an access cavity, its irrigation with 3% sodium hypochlorite (5 ml) and an application of a calcium hydroxide paste (Calcicur; Voco GmbH, Cuxhaven, Germany). The tooth was then temporary sealed using a foam pellet and a glass ionomer cement (Ketac Cem; 3M Espe, Seefeld, Germany). In the interests of patient welfare, no microbiological samples were taken at this point of the endodontic emergency treatment.”
“From this point onwards, microbiological samples were taken five times during the further treatment sequence according to the study protocol from study arm I. The root canals were therefore flooded with sterile saline solution for one minute and the microbiological samples were collected at each of the following time points using sterile paper points (ISO 30; VDW Antaeos GmbH, Munich, Germany) [2]: […]”
Reviewer: In this first emergency endodontic treatment, authors should clarify what type of temporary restoration is used (although it is written in the discussion);
Answer authors: The reviewer is right and we have checked and corrected this in the Material&Methods part:
“The tooth was then temporary sealed using a foam pellet and a glass ionomer cement (Ketac Cem; 3M Espe, Seefeld, Germany). In the interests of patient welfare, no microbiological samples were taken at this point of the endodontic emergency treatment.”
Reviewer: In the same first emergency endodontic treatment, did the teeth have all four walls necessary to start the endodontic treatment? If not, was pre-endodontic restoration performed before starting treatment?
Answer authors: Yes, the reviewer is right; we have therefore improved the description:
“The initial endodontic emergency treatment of the study participants included, if necessary, restoration of the affected tooth with an adequate pre-endodontic composite restoration to ensure a suitable initial situation for the further treatment.”
Reviewer: "2nd sampling (Root Canal Treatment)": all the root canals was instrumented with the same final file (30.09)? Even though they are molars or single-rooted teeth?
Answer authors: Thank you for this comment. The authors are happy to respond to the reviewer's statement in more detail and have adapted the manuscript as follows:
“2nd sampling [Root Canal Treatment] – after clinical screening and inclusion of appropriate teeth into the study, the included teeth were prepared via chemomechanical root canal preparation up to size 30.09 (ProTaper Gold; Dentsply Sirona GmbH, Bensheim, Germany) under irrigation with sodium hypochlorite (3%; 5 ml total, applied over the duration of the root canal preparation) and ethylenediaminetetraacetic acid (15%; 2 ml).
Reviewer: How many different operators performed the treatments? Were they calibrated with each other? How was the calibration done?
Answer authors: This is a good point by the reviewer, which we have addressed in the manuscript as follows:
“All endodontic treatment procedures were performed according to a highly standardized protocol (study-internal standardization) to ensure comparability of the results, under the guidance of three dentists experienced in endodontics who were appointed as clinical investigators within the ethics approval as well as under the responsibility of the principal investigator responsible for the study.”
Reviewer: In relation to the "Discussion" part, the authors need to modify a little bit. They should focus more on the results obtained and interpret them according to the existing scientific evidence. There is no need to evaluate topics that they have not analyzed (f. ex. single appointment vs multiple appointment). Furthermore, the discussion should focus more on calcium hydroxide (objective of the study) and not so much on irrigation with sodium hypochlorite.
Answer authors: We regret that this reviewer got a negative impression of our discussion and are therefore very pleased with his comments. We have used these to extensively revise our discussion and to better emphasize the necessity of individual points included in the discussion.
“The medicament most commonly used for temporary intracanal dressings is calcium hydroxide, which has a strong antibacterial effect due to its alkaline pH (12.5-12.8) and which is attributed to a resulting reduction of the intracanal bacterial load [6,15,16]. It is also assumed to have an influence on the outer root surface in terms of periodontal recovery [1,17,18], to be effective also against bacterial products such as lipopolysaccharides [19] and to control inflammatory exudates from the periapical area [20]. The results of the present study are largely consistent with previous findings regarding the basic assumptions about calcium hydroxide. In contrast to many previous studies, however, the results of the present study show that the total number of bacteria in the root canal increased significantly compared to the previous additional root canal irrigation (Disinfection I) despite the use of calcium hydroxide as a medicinal insert. Since the antibacterial efficacy of calcium hydroxide has previously been demonstrated in in vitro studies [4,21], the question remains as to how this renewed increase in bacterial colonization might be explained here. Considering that in vitro models of any kind can never reflect the complexity of actual in vivo situations, it must be taken into account that in vitro studies on antibacterial efficacy are often conducted using bacterial suspensions or artificial biofilms consisting of only a few bacterial species [22,23]. Therefore, also other factors, such as additional protective properties of well-established biofilms may be of decisive importance here. These properties include a biofilm matrix, an altered growth rate of biofilm organisms, as well as other physiological changes [24,25]. Thus, most bacteria also showed increased resistance to alkaline challenges/stress when organized in biofilms [24–26]. A possibly insufficient coating of the root canal wall with calcium hydroxide must also be considered here as a possible reason for a failure of the medicinal insert and the observed increase in TBL. A potential impairment of the coating of the root canal walls is the so-called vapor lock effect, which is considered a key limitation in the disinfection of root canals with rinsing solutions. This term refers to gas accumulations that usually occur in the lower third of the root canal (due to anatomical, physical or chemical influences) and prevent the deeper penetration of rinsing solutions or medicinal pastes as well as their homogeneous distribution and thus also prevent their optimal effectiveness (e.g. due to lack of contact with the inner surfaces or dentinal tubules of the root canal) [27–29]. A recent study by Puleio et al. from 2023 investigated the vapor lock phenomenon during endodontic treatment using CBCT technique and demonstrated its presence in almost all endodontic treatments, especially in the apical canal third [30]. With regard to the present study, it cannot be ruled out that the calcium hydroxide pastes used for medicinal inserts are also subjected to the phenomenon of the vapor-lock effect, thus impairing the sufficient coating of the entire surface of the root canal area. Furthermore, the ideal time that calcium hydroxide must be present in the root canal in order to comprehensively disinfect the canal system is not yet known. Nor is it known to which extent the type of bacteria as well as their location in the root canal influence the result [31]. Nevertheless, previous studies have already shown that up to 25 % of bacteria can remain within the root canal, which is consistent with the results of the present [32–34]. As a result, remaining bacteria within the root canal could proliferate despite the use of a medicinal insert and thus cause a renewed increase in the TBL value at the time of the fourth sampling [Medication]. “
Reviewer 2 Report
Comments and Suggestions for Authors
This research want to investigated the calcium hydroxide paste as a temporary root canal dressing. The topic is interesting and original in the field. The research methodology was applied correctly and carefully. All the reference are appropriates.
Some small changes are necessary in order to accept the article for publication:
1) In the "discussion" section, the role of root canal disinfection is emphasized, explaining that a dressing with calcium hydroxide could help lower the bacterial load.
One of the limitations of root canal disinfection is the formation of Vapor Lock effect, which limits the disinfection of the irrigant in the third apical root canal.
Do you think the use of calcium hydroxide can overcome this problem?
Do you think that a vapor lock effect also occurs during calcium hydroxide dressing?
I recommend adding a paragraph answering these questions about vapor lock, analyzing and citing the following article:
Puleio, F.; Lizio, A.S.; Coppini, V.; Lo Giudice, R.; Lo Giudice, G. CBCT-Based Assessment of Vapor Lock Effects on Endodontic Disinfection. Appl. Sci. 2023, 13, 9542. https://doi.org/10.3390/app13179542
2) Authors should include a paragraph explaining the clinical applications of the research.
I belive that with small modify the article could be accepted for publishing
Author Response
Dear colleague,
thank you very much for your thorough review of our manuscript.
Please find enclosed the revised version of our manuscript entitled "Temporary root canal obturation with a calcium hydroxide-based dressing: a randomized controlled clinical trial", which has been approved by all authors.
Comments and Suggestions for Authors:
Reviewer: In the "discussion" section, the role of root canal disinfection is emphasized, explaining that a dressing with calcium hydroxide could help lower the bacterial load.
One of the limitations of root canal disinfection is the formation of Vapor Lock effect, which limits the disinfection of the irrigant in the third apical root canal.
Do you think the use of calcium hydroxide can overcome this problem?
Do you think that a vapor lock effect also occurs during calcium hydroxide dressing?
I recommend adding a paragraph answering these questions about vapor lock, analyzing and citing the following article:
Puleio, F.; Lizio, A.S.; Coppini, V.; Lo Giudice, R.; Lo Giudice, G. CBCT-Based Assessment of Vapor Lock Effects on Endodontic Disinfection. Appl. Sci. 2023, 13, 9542. https://doi.org/10.3390/app1317954
Answer authors: Thank you very much for your time, effort and laud. We took this advice with great interest and read up on the literature. An interesting point, which we have included in the text as follows:
“A possibly insufficient coating of the root canal wall with calcium hydroxide must also be considered here as a possible reason for a failure of the medicinal insert and the observed increase in TBL. A potential impairment of the coating of the root canal walls is the so-called vapor lock effect, which is considered a key limitation in the disinfection of root canals with rinsing solutions. This term refers to gas accumulations that usually occur in the lower third of the root canal (due to anatomical, physical or chemical influences) and prevent the deeper penetration of rinsing solutions or medicinal pastes as well as their homogeneous distribution and thus also prevent their optimal effectiveness (e.g. due to lack of contact with the inner surfaces or dentinal tubules of the root canal) [27–29]. A recent study by Puleio et al. from 2023 investigated the vapor lock phenomenon during endodontic treatment using CBCT technique and demonstrated its presence in almost all endodontic treatments, especially in the apical canal third [30]. With regard to the present study, it cannot be ruled out that the calcium hydroxide pastes used for medicinal inserts are also subjected to the phenomenon of the vapor-lock effect, thus impairing the sufficient coating of the entire surface of the root canal area. Furthermore, the ideal time that calcium hydroxide must be present in the root canal in order to comprehensively disinfect the canal system is not yet known. Nor is it known to which extent the type of bacteria as well as their location in the root canal influence the result [31]. Nevertheless, previous studies have already shown that up to 25 % of bacteria can remain within the root canal, which is consistent with the results of the present [32–34]. As a result, remaining bacteria within the root canal could proliferate despite the use of a medicinal insert and thus cause a renewed increase in the TBL value at the time of the fourth sampling [Medication].”
Reviewer: Authors should include a paragraph explaining the clinical applications of the research.
Answer authors: Many thanks for this comment. We took this as an opportunity to make some changes to the discussion section of the manuscript and to better address our points worth discussing:
“The medicament most commonly used for temporary intracanal dressings is calcium hydroxide, which has a strong antibacterial effect due to its alkaline pH (12.5-12.8) and which is attributed to a resulting reduction of the intracanal bacterial load [6,15,16]. It is also assumed to have an influence on the outer root surface in terms of periodontal recovery [1,17,18], to be effective also against bacterial products such as lipopolysaccharides [19] and to control inflammatory exudates from the periapical area [20]. The results of the present study are largely consistent with previous findings regarding the basic assumptions about calcium hydroxide. In contrast to many previous studies, however, the results of the present study show that the total number of bacteria in the root canal increased significantly compared to the previous additional root canal irrigation (Disinfection I) despite the use of calcium hydroxide as a medicinal insert. Since the antibacterial efficacy of calcium hydroxide has previously been demonstrated in in vitro studies [4,21], the question remains as to how this renewed increase in bacterial colonization might be explained here. Considering that in vitro models of any kind can never reflect the complexity of actual in vivo situations, it must be taken into account that in vitro studies on antibacterial efficacy are often conducted using bacterial suspensions or artificial biofilms consisting of only a few bacterial species [22,23]. Therefore, also other factors, such as additional protective properties of well-established biofilms may be of decisive importance here. These properties include a biofilm matrix, an altered growth rate of biofilm organisms, as well as other physiological changes [24,25]. Thus, most bacteria also showed increased resistance to alkaline challenges/stress when organized in biofilms [24–26]. “
“With regard to the clinical significance of the present study, it should be noted that the observation period of the calcium hydroxide insert is limited to only one week and that no date on the overall (clinical and radiological) success of the endodontic treatments were included. Particularly with regard to the clinically relevant long-term success of endodontic treatments, it would be interesting to evaluate see the follow-up of the patients treated in the scope of this study – possibly by means of a retrospective study. Nevertheless, within the limitations of this study, the results may support previous assumptions that successful root canal treatment depends on keeping the bacterial count below a threshold that the immune system can cope with. The long-term success of root canal therapy, however, cannot always be guaranteed due to a resurgence of bacteria in cases of immune suppression brought on by illness or, for example, aging [50]. Endodontic therapy should therefore always focus on the greatest possible reduction in bacteria. In addition to chemomechanical preparation and adjuvant disinfection protocols, which are generally and still considered to be the most important step in root canal disinfection, the introduction of intracanal medications such as calcium hydroxide is considered necessary in order to keep the bactericidal effects of the irrigation solutions constant and in addition to achieve maximum eradication of pathogens from the root canal [9,15,41,51].”
Round 2
Reviewer 1 Report
Comments and Suggestions for Authors
In my opinion, the manuscript has the necessary conditions for publication.